# Improving maternal and newborn health services in Northeast Nigeria through a government-led partnership of stakeholders: a quasi-experimental study

Barbara Willey ![ORCID],[1] Nasir Umar ![ORCID],[1] Emma Beaumont ![ORCID],[1] Elizabeth Allen ![ORCID],[1] Jennifer Anyanti,[2] Abubakar Bala Bello,[3] Antoinette Bhattacharya ![ORCID],[1] Josephine Exley ![ORCID],[1] Krystyna Makowiecka ![ORCID],[1] Magdalene Okolo,[2] Rabi Sani,[3] Joanna Schellenberg ![ORCID],[1] Neil Spicer ![ORCID],[1] Umar Adamu Usman,[4] Ahmed Mohammed Gana,[5] Abdulrahman Shuaibu,[6] Tanya Marchant ![ORCID][1]

For numbered affiliations see end of article.

**Correspondence to**
Dr Barbara Willey;
barbara.willey@lshtm.ac.uk

## ABSTRACT

**Objectives** This study aimed to quantify change in the coverage, quality and equity of essential maternal and newborn healthcare interventions in Gombe state, Northeast Nigeria, following a four year, government-led, maternal and newborn health intervention.

**Design** Quasi-experimental plausibility study. Repeat cross-sectional household and linked health facility surveys were implemented in intervention and comparison areas.

**Setting** Gombe state, Northeast Nigeria.

**Participants** Each household survey included a sample of 1000 women aged 13–49 years with a live birth in the previous 12 months. Health facility surveys comprised a readiness assessment and birth attendant interview.

**Interventions** Between 2016–2019 a complex package of evidence-based interventions was implemented to increase access, use and quality of maternal and newborn healthcare, spanning the six WHO health system building blocks.

**Outcome measures** Eighteen indicators of maternal and newborn healthcare.

**Results** Between 2016 and 2019, the coverage of all indicators improved in intervention areas, with the exception of postnatal and postpartum contacts, which remained below 15%. Greater improvements were observed in intervention than comparison areas for eight indicators, including coverage of at least one antenatal visit (71% (95% CI 62 to 68) to 88% (95% CI 82 to 93)), at least four antenatal visits (46% (95% CI 39 to 53) to 69% (95% CI 60 to 75)), facility birth (48% (95% CI 37 to 59) to 64% (95% CI 54 to 73)), administration of uterotonics (44% (95% CI 34 to 54) to 59% (95% CI 50 to 67)), delayed newborn bathing (44% (95% CI 36 to 52) to 62% (95% CI 52 to 71)) and clean cord care (42% (95% CI 34 to 49) to 73% (95% CI 66 to 79)). Wide-spread inequities persisted however; only at least one antenatal visit saw pro-poor improvement.

**Conclusions** This intervention achieved improvements in life-saving behaviours for mothers and newborns,

## Strengths and limitations of this study

► This was a large, multiyear study.
► Tools measured access, use, quality and equity of maternal and newborn healthcare in a setting with a high mortality burden.
► Standardised tools were applied each year to collect both population level and facility level evidence.
► The selection of intervention and comparison areas was pragmatic, following government implementation plans.
► In this non-randomised design, baseline characteristics between areas were not balanced, and residual confounding by area cannot be discounted.

demonstrating that multipartner action, coordinated through government leadership, can shift the needle in the right direction, even in resource-constrained settings.

## INTRODUCTION

Interventions to reduce maternal and newborn morbidity and mortality are available,[1–6] however, implementation gaps create a mismatch between the efficacy and effectiveness of interventions at a population level.[7–9] Despite substantial progress in some regions during the era of the Millennium Development Goals,[10] maternal and neonatal mortality remain unacceptably high, especially in low-income and middle-income countries.[11–13] The era of Universal Health Coverage highlights the need for a focus on quality of care and reducing inequalities alongside increasing access to care.[14–19]

Challenges remain in how to equitably scale and package efficacious interventions

**Figure 1** Intervention components by health system strengthening building block. CHEW, community health extension worker; HMIS, Health Management Information System; HSS, health system strengthening; MNH, maternal and newborn health; MPDSR, Maternal Perinatal Death Surveillance and Response; PHC, primary health centre; VHW, village health worker.

and integrate them within existing health systems.[20] Examining maternal, newborn and child health coverage data from 36 sub-Saharan African countries, Wehrmeister and colleagues concluded that countries had narrowed equity gaps since 1995, although wealth-related inequalities remain highest in West Africa.[21] In their study of 83 low-income and middle-income countries, Barros and colleagues emphasised the need for implementation of pro-poor strategies if the most vulnerable are to benefit from available healthcare.[22] Part of the solution to these challenges may be robust governance and government-led initiatives that focus on universal and equitable access.[23 24]

Here we report the effect of the Gombe Partnership for Maternal and Newborn Health, a government-led intervention implemented between 2016 and 2019 in Northeast Nigeria and designed to coordinate multiple actors toward the goal of equitably improving high quality maternal and newborn health services. An intended outcome of the partnership was to inform scale up and share learning across neighbouring states and regions. Implementation used an adaptive management approach and partners met regularly to examine monitoring data and amend implementation as necessary. In this paper, we examine the effect of this intervention on the coverage and quality of essential maternal and newborn healthcare interventions after four years of implementation, and assess whether any improvements were equitable across socioeconomic groups.

## METHODS
### Setting and intervention implementation
This study was conducted in Gombe state, a predominately rural (80%) and sparsely-populated state in Northeast Nigeria,[25] where the burden of maternal and neonatal mortality is higher than the national average and stood at 1549 maternal deaths per 100 000 live births in 2015 and 33 neonatal deaths per 1000 live births in 2017.[26 27] In 2016, when this study started, 29% of women gave birth in a health facility,[25] principally at primary health centres (PHC).[28 29] Primary healthcare services

are predominately delivered by the Gombe State Primary Health Care Development Agency, with little private sector provision, unlike some other regions of Nigeria.[30] In 2016, 460 PHCs in the state provided antenatal care (ANC), birth and intrapartum services,[28] mainly delivered by community health extension workers (CHEWs), junior CHEWs and community health officers, but very few nurses, midwives or doctors.[31]

From 2016 to 2019, the government led a maternal and newborn health partnership to improve access, use and quality of maternal and newborn health services. Within this partnership, non-governmental organisations (NGOs) implemented a package of evidence-based interventions that spanned the six WHO health system building blocks[32] (figure 1). Components aimed to enhance uptake and provision of life-saving interventions at three interacting levels (individuals and families; community organisations; and the health system). At the individual and family level, interventions aimed to improve knowledge, attitudes and practices to increase enhanced home-based practices and increase demand for routine professional care; for example, a community-based village health worker home visit scheme was initiated to improve knowledge about and linkages between families and health services.[33] At the community organisation level, interventions aimed to improve trust and accountability between the family and health system levels; for example, through supporting community-based mothers groups to interact with their local primary health services.[34] Interventions at the health system level aimed to improve the supply of safe, effective and high quality care; for example, working with government to strengthen the supply chain for essential drugs in PHCs. Underpinning these three levels of engagement were interventions designed to raise public awareness about maternal and newborn health across the state through mass media and advocacy events.[35]

This package of interventions was deliberately coordinated by government as a pathway towards improved maternal and newborn outcomes.[36] The government, NGOs, partners and the funder met every six months to review monitoring data, trouble-shoot implementation challenges, course-correct and reinvigorate communal purpose towards a shared goal. To facilitate learning, the package of interventions was initially implemented in an intervention area, with a view to scaling-up to the entire state. The intervention area was defined as 57 subdistrict level wards (half of the state's 114 wards), purposively identified by government. Community-based demand generation activities were implemented in these wards, and one centrally located PHC within each ward was chosen to implement the activities designed to improve health service quality: the rationale being that it was preferable to have one well-functioning PHC per ward, rather than a larger number of less well-functioning facilities. Residents of the state's remaining 57 wards (comparison area) continued to receive their usual care, with the exception of mass media components which were state-wide (figure 1).

## Study design

We used a quasi-experimental plausibility study design[37] to explore the association between the intervention and indicators of use and quality of maternal and newborn health services, comparing changes observed over time in intervention areas to those in comparison areas. Repeat cross-sectional household and linked health facility surveys were undertaken. Survey methods were replicated each time. Data collection tools were informed by existing large scale survey tools such as the Demographic and Health Surveys[38] and Service Availability and Readiness Assessment Surveys.[39]

We conducted four annual household surveys during July/August 2016–2019. The household survey consisted of a modular household questionnaire: (1) A household module which asked about characteristics of the household and ownership of commodities as proxy markers of household socioeconomic status, and during which a household roster of all usually resident people was generated; (2) A women's module which asked all resident women about the healthcare available to them, their recent contact with health services and their recent birth history; and (3) A mother's module which asked all women who reported a birth in the last 12 months about their contact with health services across the continuum of care from pregnancy to postnatal care. In a household with a resident recently delivered woman the questionnaire took approximately 90 minutes to complete.

A random sample of 80 clusters was selected: 40 each from intervention and comparison wards. Clusters were segmented enumeration areas as defined by the National Population Commission. Within each cluster, all households were visited. During each survey, we aimed to survey a total of 6000 households across the 80 clusters. This was expected to result in interviews with 1000 women with a live birth in the previous 12 months (ie, 500 each in intervention and comparison). Sample size calculations assumed a design effect of 2.5, 95% probability and 80% power. For each survey, where comparison prevalence of indicators ranged between 20% and 60%, this sample size was sufficient to detect changes of 15 percentage points in intervention areas.

During the same period, we conducted seven facility and birth attendant surveys at six monthly intervals. Four surveys were done concurrently with the household surveys

(figure 2). At each time point, a facility readiness assessment was carried out plus an interview was conducted with the available birth attendant who had attended the most recent delivery recorded in the maternity register. The facility survey took approximately 120 minutes to complete all sections. All 57 intervention PHCs were surveyed plus one PHC selected at random from each of the 40 comparison clusters sampled during household surveys. For each survey, where prevalence of indicators in comparison facilities ranged between 5% and 70%, this sample of 97 facilities was sufficient to detect changes of 20 percentage points in intervention areas.

All recruited interviewers were from Gombe state and attended a one week training course at each survey. Eight survey teams were recruited in total including a supervisor, four household interviewers, one facility and birth attendant interviewer, one mapper who listed households and segmented enumeration areas as necessary and one data support member. Questionnaires were translated and back-translated between English and Hausa languages to ensure consistency and were pretested. As part of the week long training, the full study protocol was pilot tested in two clusters to identify and correct any operational or language problems. All data were collected using hand-held digital devices and synchronised with a supervisor laptop each day. Automated summary reports were produced to identify and address any internal inconsistencies.

## Statistical analysis of effectiveness and equity

Eighteen indicators measuring the coverage of life-saving commodities and behaviours were selected a priori for their known association with maternal and newborn health outcomes (table 1). Changes over the four year study period were compared between intervention and comparison areas. Analyses were at the individual level for household survey indicators and at the health facility level for facility and birth attendant indicators. Data were analysed in Stata V.15 (StataCorp, 2017).

Household and health facility characteristics were summarised using appropriate summary statistics. We used survey commands (*svy*) to account for clustering. We identified variables which differed significantly between intervention and comparison areas in 2016 using a design-based F test and included their cluster-level means in regression models. Percentage point differences and 95% confidence intervals (CIs) in indicator coverage between 2016 and 2019 were calculated.

Effectiveness of the programme on pre-specified indicators was estimated through logistic regression models. A likelihood ratio test comparing models with time as a continuous variable to one where each time point was included separately, determined whether time was included as a continuous or categorical variable. Models included fixed effects for area (intervention vs comparison), time (at all time points) and the interaction between area and time, to describe any additional effect in 2019 in the intervention areas compared with

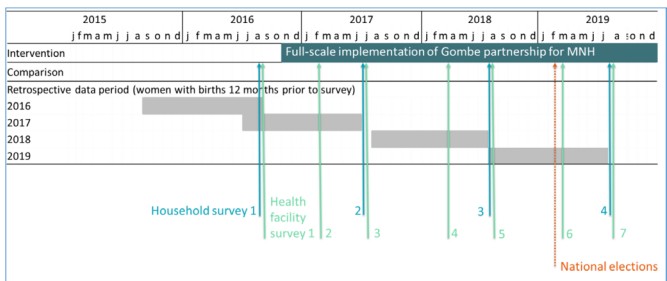

**Figure 2** Study timeline. MNH, maternal and newborn health.

**Table 1** Indicator definitions

| Indicator name | Numerator | Denominator |
|---|---|---|
| Antenatal care period | | |
| Women made at least one ANC visit | Number of women aged 15–49 years with a birth during 12 months before survey who made at least one ANC visit | Women aged 15–49 years with a birth during 12 months before survey |
| Women made at least four ANC visits | Number of women aged 15–49 years with a birth during 12 months before survey who made at least four ANC visits | Women aged 15–49 years with a birth during 12 months before survey |
| Women received basic ANC quality | Number of women aged 15–49 years with a birth during 12 months before survey attending ANC who received all three of: urine test, blood test and blood pressure measurement | Women aged 15–49 years with a birth during 12 months before survey |
| Women received syphilis test results | Number of women aged 15–49 years with a birth during 12 months before survey who received a test result for syphilis | Women aged 15–49 years with a birth during 12 months before survey |
| Facilities had syphilis treatment available | Number of primary health facilities which had benzathine penicillin for syphilis treatment in stock | Primary health facilities surveyed |
| Intrapartum care period | | |
| Facility delivery | Number of women aged 15–49 years with a birth during 12 months before survey who gave birth in a health facility | Women aged 15–49 years with a birth during 12 months before survey |
| Uterotonics | Number of women aged 15–49 years with a birth during 12 months before survey who received prophylactic uterotonics, includes facility and home births | Women aged 15–49 years with a birth during 12 months before survey |
| Facilities had pre-eclampsia treatment available | Number of primary health facilities which had magnesium sulphate available for treatment of pre-eclampsia | Primary health facilities surveyed |
| Birth attendant was able to correctly manage pre-eclampsia | Number of birth attendants who were able to correctly manage pre-eclampsia, as assessed by clinical vignette | Birth attendants interviewed |
| Immediate newborn care period | | |
| Facilities had newborn resuscitation equipment available | Number of primary health facilities which had a bag and mask for newborn resuscitation (size 0 and size 1) | Primary health facilities surveyed |
| Birth attendant was able to correctly manage newborn resuscitation | Number of birth attendants who were able to correctly manage neonatal resuscitation, as assessed by simulation | Birth attendants interviewed |
| Newborn bathing was delayed until at least 24 hours after birth | Number of newborns born to women aged 15–49 years with a birth during 12 months before survey who had delayed bathing for the first 24 hours of life | Women aged 15–49 years with a birth during 12 months before survey |
| Newborn received clean cord care | Number of newborns born to women aged 15–49 years with a birth during 12 months before survey who received clean cord care, defined as cutting cord with clean blade or scissors, administering chlorhexidine to the newborn cord and not administering anything harmful to the newborn cord | Women aged 15–49 years with a birth during 12 months before survey |
| Newborn was breast fed within 1 hour of birth | Number of newborns born to women aged 15–49 years with a birth during 12 months before survey for which breast feeding was initiated within 1 hour of birth | Women aged 15–49 years with a birth during 12 months before survey |
| Postpartum and postnatal care period | | |
| Women received a postpartum check within 2 days of delivery | Number of women aged 15–49 years with a birth during 12 months before survey who had a postpartum check-up within 2 days of delivery | Women aged 15–49 years with a birth during 12 months before survey |
| Newborns received a postnatal check within 2 days of birth | Number of newborns born to women aged 15–49 years with a birth during 12 months before survey who had a postnatal check-up within 2 days of birth | Women aged 15–49 years with a birth during 12 months before survey |
| Newborns with suspected sepsis treated with antibiotics | Number of newborns aged 0–60 days with suspected sepsis who received antibiotics | Newborns aged 0–60 days with suspected sepsis |
| Facilities had newborn sepsis treatment available | Number of primary health facilities with amoxicillin and gentamicin available | Primary health facilities surveyed |

Data were collected for women aged 13–49 years. However, no women <15 years of age had a live birth in the 12 months before the survey.
ANC, antenatal care.

2016 in the comparison areas. Cluster level means of variables which differed between intervention and comparison areas in 2016, identified using a design-based F-test, were included in all models. We included a random effect for cluster to account for clustering in logistic regression models of household survey indicators.

To examine equity, principal components analysis was used to generate an index of household wealth, based on asset ownership.[36] Using this, households were categorised into quintiles from poorest to least poor. Where there was evidence of greater improvement in the intervention areas compared with the comparison areas, we

**Table 2** Interviews conducted, by survey and year

| | 2016 | 2017 | 2018 | 2019 |
|---|---|---|---|---|
| Number of clusters | 80 | 80 | 80 | 80 |
| **Households** | | | | |
| Target number of households to be surveyed | 6000 | 6000 | 6000 | 6000 |
| Number of households surveyed | 5747 | 5762 | 5925 | 5616 |
| Response rate | 96% | 96% | 99% | 94% |
| **Resident women** | | | | |
| Number of resident women aged 15–49 years | 8556 | 8395 | 10177 | 10080 |
| Number of women interviewed | 8453 | 8270 | 10150 | 9959 |
| Response rate | 99% | 98% | 99% | 99% |
| Number of resident women with live birth in 12 months preceding survey | 1011 | 932 | 991 | 871 |
| **Primary health facilities** | | | | |
| Target number of primary health facilities to be surveyed | 97 | 97 | 97 | 97 |
| Number of primary health facilities surveyed | 94 | 94 | 97 | 97 |
| Response rate | 97% | 97% | 100% | 100% |
| Number of birth attendants interviewed (one per facility) | 92 | 91 | 93 | 95 |
| Response rate | 95% | 94% | 96% | 98% |

Note: The survey sampled resident women aged 13–49 years, however, no births in the last 12 months were recorded among women aged 13–14 years and as such the sample of women with a recent birth is aged 15–49 years.

examined the difference of change over time by household socioeconomic quintile in the intervention areas. We tested for interaction between time and socioeconomic status quintile.

### Patient and public involvement

Patients and the public were not involved in the design, conduct, reporting or dissemination plans of our research.

### RESULTS
### Sample description

Table 2 shows the number of households, resident women and facilities surveyed. High response rates were seen throughout. Table 3 presents baseline characteristics for the 1011 women interviewed with a recent live birth. Women had a mean age of 26 years (Standard Deviation (SD) 0.4) and parity of four births (SD 0.2); over one-quarter reported more than six births. Marriage was near universal (>94%) and 82% of women were Muslim. Mean year of schooling was 4.3 (SD 0.6) in intervention and 2.6 years (SD 0.5) in comparison (p=0.06). Although the majority reported no formal education, this was substantially higher in comparison areas (72% vs 55%, p=0.05).

Table 4 shows that in 2016, almost all sampled facilities provided ANC and delivery services, and at least 70% of facilities provided postnatal care. Only a small minority of facilities had a trained midwife available 24 hours a day. In total, a third of intervention PHCs, and a quarter of comparison PHCs provided all basic emergency obstetrical and newborn care (BEmONC) signal functions (excluding assisted vaginal delivery) in the

previous 3 months (p=0.58). Facilities had low volumes of deliveries (<500 births per year),[40] with each intervention PHC managing a mean of 288 births per year versus 192 in comparison PHCs. In intervention PHCs 55 birth attendant were interviewed, and 37 in comparison PHCs. The cadre of the interviewed birth attendant was similar between intervention and comparison PHCs. Very few were nurses or midwives (4% in intervention and none in comparison), and approximately half in both areas were community health extension workers, while the remaining 50% in each area were non-clinical health workers. These birth attendants were similar in age, a mean of 36 years (SD 1.0) in intervention and of 39 years (SD 1.3) in comparison PHCs, and both had an average of 13 years' education (SD 0.6). On average, duration in post was 5 years (SD 0.6) in intervention and 4 years (SD 0.7) in comparison PHCs, suggesting relatively stable staff turnover.

### Change over time in intervention and comparison areas

Table 5 shows prevalence of 18 maternal and newborn health indicators between 2016 and 2019 in intervention and comparison areas for women with a recent live birth, PHCs and surveyed birth attendants. With the exception of postpartum and postnatal health checks, which did not improve, 16 indicators improved over time in the intervention area (table 5).

Table 5 shows that seven indicators improved in the intervention areas only, while nine also improved in comparison areas. For eight indicators, greater changes over time were observed in intervention areas above any

**Table 3** Baseline characteristics: women with recent birth (2016 survey)

| | Intervention | | Comparison | | |
|---|---|---|---|---|---|
| | N=519 | % (95% CI) | N=492 | % (95% CI) | P value† |
| **Woman's age category** | | | | | |
| 15–19 years | 54 | 10 (8 to 14) | 60 | 12 (9 to 16) | 0.35 |
| 20–24 years | 163 | 32 (27 to 37) | 128 | 26 (23 to 30) | |
| 25–29 years | 146 | 28 (25 to 32) | 128 | 26 (22 to 31) | |
| 30–34 years | 87 | 17 (14 to 20) | 94 | 19 (16 to 23) | |
| 35–39 years | 48 | 9 (7 to 12) | 56 | 11 (9 to 14) | |
| 40–44 years | 15 | 3 (2 to 5) | 21 | 4 (3 to 7) | |
| 45–49 years | 3 | 0.6 (0.2 to 2) | 4 | 0.8 (0.3 to 2) | |
| **Parity** | | | | | |
| 1 birth | 111 | 21 (18 to 26) | 79 | 16 (13 to 20) | 0.07 |
| 2 births | 76 | 15 (11 to 19) | 83 | 17 (14 to 20) | |
| 3–5 births | 205 | 39 (35 to 44) | 178 | 37 (32 to 41) | |
| ≥6 births | 127 | 25 (20 to 30) | 152 | 31 (27 to 35) | |
| **Marital status** | | | | | |
| Currently married | 488 | 94 (89 to 97) | 472 | 96 (93 to 98) | 0.35 |
| Not married | 31 | 6 (3 to 10) | 20 | 4 (2 to 7) | |
| **Religion** | | | | | |
| Christianity | 92 | 18 (10 to 31) | 71 | 14 (8 to 25) | 0.62 |
| Islam | 426 | 82 (69 to 90) | 421 | 86 (75 to 92) | |
| **Education level completed** | | | | | |
| None | 283 | 55 (42 to 67) | 353 | 72 (60 to 81) | 0.05 |
| 1–7 years (primary) | 87 | 17 (11 to 24) | 56 | 11 (7 to 19) | |
| ≥8 years (secondary) | 149 | 29 (20 to 39) | 83 | 17 (11 to 26) | |
| **Household socioeconomic status, by quintile*** | | | | | |
| Quintile 1 (poorest) | 102 | 20 (14 to 27) | 113 | 23 (15 to 33) | 0.44 |
| Quintile 2 | 96 | 19 (14 to 24) | 111 | 23 (17 to 29) | |
| Quintile 3 | 114 | 22 (17 to 27) | 96 | 20 (15 to 24) | |
| Quintile 4 | 94 | 18 (14 to 24) | 100 | 20 (15 to 27) | |
| Quintile 5 (richest) | 113 | 22 (14 to 33) | 72 | 15 (9 to 23) | |

*Socioeconomic asset index: wall material not thatch or mud, floor material not earth or sand, clean source of drinking water, roof material iron, tiles or cement, flush toilet or latrine, electricity supply to home, ownership of fridge, television, radio, bike, kerosene lamp, wrist watch, motorcycle, generator, fan.
†P value from design-based F test.
CI, Confidence interval.

increases seen in comparison areas. The proportion of women attending ANC visits and giving birth in a facility increased to 88% for ANC1, 69% for ANC4 and 64% for facility birth. These represented improvements up to nine percentage points above increases seen in comparison areas (p=0.006 and p=0.003, respectively for ANC1 and ANC4, and p<0.001 for facility birth). For the intrapartum period, we saw an increase in the administration of prophylactic uterotonics among all women reaching 59%, a four percentage point improvement above changes in the comparison areas (p=0.008). Availability of treatment for pre-eclampsia increased by 32 percentage points above changes in the comparison areas (p=0.021). Adoption of life-saving behaviours for immediate newborn care also increased, with improvements up to 10 percentage points higher than changes seen in comparison areas (p=0.034 for delayed bathing of the newborn, and p<0.001 for clean cord care, respectively). Availability of newborn bag and mask for resuscitation increased by 36 percentage points above changes in the comparison areas (p=0.001). Similar improvements were observed in intervention and comparison areas for three indicators: women receiving syphilis test results (p=0.758); birth attendants' capability to manage pre-eclampsia appropriately (p=0.971)

**Table 4** Baseline characteristics: health facilities (2016 survey)

| | Intervention | | Comparison | | |
|---|---|---|---|---|---|
| | N=55 | (%, 95% CI) | N=39 | (%, 95% CI) | P value* |
| *Infrastructure* | | | | | |
| Clean source of running water | 34 | 62 (48 to 74) | 17 | 44 (29 to 60) | 0.09 |
| Electricity connection | 38 | 69 (56 to 80) | 24 | 62 (45 to 76) | 0.45 |
| *Opening hours* | | | | | |
| 7 days a week | 44 | 80 (67 to 89) | 25 | 68 (51 to 81) | 0.18 |
| *Services offered* | | | | | |
| ANC services available | 55 | 100 | 37 | 95 (81 to 99) | 0.09 |
| Delivery services available | 55 | 100 | 34 | 87 (72 to 95) | 0.01 |
| Postnatal care services available | 42 | 76 (63 to 86) | 27 | 69 (53 to 82) | 0.45 |
| 24-hour delivery services available 7 days a week | 44 | 83 (70 to 91) | 27 | 82 (65 to 92) | 0.08 |
| Trained midwife available 24 hours a day 7 days a week | 14 | 15 (9 to 24) | 12 | 13 (7 to 21) | 0.57 |
| Provided all BEmONC signal functions (excluding assisted vaginal delivery) in previous 3 months | 17 | 31 (20 to 45) | 10 | 26 (14 to 42) | 0.58 |

*P value from design-based F test.

ANC, antenatal care; BEmONC, basic emergency obstetrical and newborn care; CI, Confidence interval.

and immediate breast feeding of the newborn (p=0.524) (table 5).

### Change in coverage 2016–2019 by household wealth quintile

Six indicators from the household survey data demonstrated population level coverage improvement over time and were included in the equity analysis (figure 3). Five of these were inequitable at baseline, with lower coverage among the poorest households. By 2019, although the change over time in coverage was even across all quintiles, baseline patterns of inequity persisted for three indicators: women attending at least four ANC visits (p=0.095), administration of uterotonics (p=0.118) and delayed bathing of the newborn (p=0.671). The improvement over time was inequitable for two indicators, with women in the poorest households benefiting the least: clean cord care (p=0.020) and facility birth (p=0.005). By contrast, improvement over time for at least one ANC visit was pro-poor (p=0.055), with coverage improving by 17 percentage points among the poorest households (from 58% to 75%), and by five percentage points to 91% among women in the least-poor households.

### DISCUSSION

Over a four year period in this rural and resource-constrained setting, this government-led maternal and newborn health partnership was able to improve several high impact indicators over and above any background changes observed in comparison areas. However, there

was a marked lack of improvement in timely postnatal or postpartum care, coverage of which remained below 15%. Additionally, despite improvements in coverage indicators, socioeconomic inequalities in the use of maternal and newborn health services generally persisted.

Multiple improvement efforts were implemented simultaneously by the partnership. For example, increases in ANC visits and facility births were supported by more than one implementing partner. This reinforcement enabled the utilisation coverage for these services to improve, even when historical data demonstrated little change during previous initiatives.[41 42] Changes to quality of care were mixed. We saw substantial increases in input quality where implementing partner actions directly strengthened pre-existing systems, for example, where NGOs strengthened and managed existing supply chains for medicines and commodities and returned management to the government in a phased manner. However, this success was not replicated with process quality indicators. There was no excess improvement in process indicators of quality ANC, such as measuring blood pressure and taking a urine and blood sample during an ANC visit. Staffing and workload were especially challenging and it was not uncommon for one staff member to be attending to multiple women and babies.[43] The challenge of changing health worker behaviour is well-established in the literature.[44 45] Challenges in providing complete and timely care have been noted in other low-income and middle-income settings.[46]

**Table 5** Indicators over time by intervention and comparison area

| Indicator | Area | 2016 | | | 2017 | | | 2018 | | | 2019 | | | Percentage point difference 2019–2016 (95% CI) P value* | Linear trend P value† | P value interaction time and area‡ |
|---|---|---|---|---|---|---|---|---|---|---|---|---|---|---|---|---|
| | | N | n | % (95% CI) | N | n | % (95% CI) | N | n | % (95% CI) | N | n | % (95% CI) | | | |
| **Antenatal care period** | | | | | | | | | | | | | | | | |
| Women made at least one ANC visit | Intervention | 519 | 366 | 71 (62 to 78) | 494 | 350 | 71 (64 to 77) | 511 | 435 | 85(79 to 90) | 458 | 405 | 88(82 to 93) | 17 (11 to 25) <0.001 | <0.001 | 0.006 |
| | Comparison | 492 | 279 | 57 (47 to 66) | 438 | 239 | 55 (46 to 63) | 480 | 292 | 61 (51 to 70) | 413 | 298 | 72(61 to 81) | 15 (6 to 25) 0.002 | 0.014 | |
| Women made at least four ANC visits | Intervention | 519 | 239 | 46 (39 to 53) | 494 | 230 | 46 (39 to 53) | 511 | 326 | 64 (56 to 71) | 511 | 315 | 69 (60 to 76) | 23 (14 to 32) <0.001 | 0.002 | 0.003 |
| | Comparison | 493 | 145 | 29 (23 to 37) | 438 | 131 | 30 (23 to 37) | 480 | 164 | 34 (25 to 45) | 480 | 179 | 43 (34 to 53) | 14 (6 to 22) <0.001 | 0.458 | |
| Women received basic ANC quality | Intervention | 519 | 303 | 58 (50 to 66) | 494 | 261 | 53 (44 to 62) | 511 | 312 | 61 (52 to 69) | 458 | 305 | 66 (58 to 74) | 8 (1 to 18) 0.008 | 0.002 | 0.599 |
| | Comparison | 492 | 216 | 44 (35 to 53) | 438 | 160 | 37 (29 to 45) | 480 | 183 | 38 (30 to 47) | 413 | 220 | 53 (42 to64) | 9 (−0.2 to 19) 0.056 | NA | |
| Women received syphilis test results | Intervention | 519 | 61 | 12 (9 to 16) | 494 | 86 | 17 (12 to 24) | 511 | 166 | 32 (26 to 40) | 458 | 149 | 33 (27 to 38) | 21 (14 to 27) <0.001 | <0.001 | 0.758 |
| | Comparison | 492 | 39 | 8 (6 to 11) | 438 | 55 | 13 (8 to 20) | 480 | 96 | 20 (14 to 27) | 413 | 88 | 21 (15 to 30) | 13 (5 to 21) 0.002 | 0.008 | |
| Facilities PHC had syphilis treatment available | Intervention | 55 | 5 | 9 (4 to 20) | 55 | 29 | 53 (39 to 66) | 57 | 37 | 65 (52 to 76) | 57 | 21 | 37 (25 to 50) | 28 (12 to 44) <0.001 | <0.001 | 0.061 |
| | Comparison | 39 | 3 | 8 (2 to 22) | 39 | 7 | 18 (9 to 33) | 40 | 6 | 15 (7 to 30) | 41 | 5 | 12 (5 to 26) | 4 (−8 to 17) 0.467 | NA | |
| **Intrapartum care period** | | | | | | | | | | | | | | | | |
| Women gave birth in a facility | Intervention | 519 | 248 | 48 (37 to 59) | 494 | 217 | 44 (33 to 55) | 511 | 313 | 61 (53 to 69) | 458 | 294 | 64 (54 to 73) | 16 (7 to 26) 0.002 | <0.001 | <0.001 |
| | Comparison | 492 | 128 | 26 (18 to 35) | 436 | 154 | 35 (27 to 45) | 480 | 147 | 31 (23 to 39) | 412 | 145 | 35 (26 to 46) | 9 (0.5 to 18) 0.039 | 0.078 | |
| Women received uterotonic | Intervention | 509 | 223 | 44 (34 to 54) | 473 | 186 | 39 (30 to 49) | 505 | 264 | 52 (45 to 60) | 435 | 256 | 59 (50 to 67) | 15 (6 to 24) 0.001 | 0.002 | 0.008 |
| | Comparison | 487 | 133 | 27 (20 to 37) | 416 | 127 | 31 (23 to 40) | 475 | 132 | 28 (21 to 35) | 372 | 141 | 38 (29 to 48) | 11 (3 to 19) 0.010 | 0.527 | |
| PHC had pre-eclampsia treatment available | Intervention | 55 | 11 | 20 (11 to 33) | 55 | 35 | 64 (51 to 75) | 57 | 56 | 98 (88 to 100) | 57 | 53 | 93 (82 to 97) | 73 (60 to 86) <0.001 | <0.001 | 0.021 |
| | Comparison | 39 | 5 | 13 (5 to 28) | 39 | 7 | 18 (9 to 33) | 40 | 23 | 58 (42 to 72) | 41 | 22 | 54 (38 to 68) | 41 (24 to 58) <0.001 | 0.062 | |

Continued

**Table 5** Continued

| Indicator | Area | 2016 | | | 2017 | | | 2018 | | | 2019 | | | Percentage point difference 2019–2016 (95% CI) P value* | Linear trend P value† | P value interaction time and area‡ |
|---|---|---|---|---|---|---|---|---|---|---|---|---|---|---|---|---|
| | | N | n | % (95% CI) | N | n | % (95% CI) | N | n | % (95% CI) | N | n | % (95% CI) | | | |
| Birth attendant was able to correctly manage pre-eclampsia | Intervention | 55 | 16 | 29 (18 to 43) | 55 | 23 | 42 (29 to 55) | 57 | 46 | 81 (68 to 89) | 57 | 50 | 88 (76 to 94) | 59 (42 to 75) <0.001 | 0.023 | 0.971 |
| | Comparison | 37 | 10 | 27 (15 to 44) | 36 | 11 | 31 (18 to 48) | 36 | 19 | 53 (36 to 69) | 38 | 27 | 71 (55 to 83) | 44 (24 to 64) <0.001 | 0.005 | |
| **Immediate newborn care period** | | | | | | | | | | | | | | | | |
| Facilities had newborn resuscitation equipment available | Intervention | 55 | 35 | 64 (50 to 75) | 55 | 45 | 82 (69 to 90) | 57 | 51 | 89 (78 to 95) | 57 | 54 | 95 (85 to 98) | 31 (17 to 46) <0.001 | <0.001 | 0.001 |
| | Comparison | 39 | 17 | 44 (29 to 60) | 39 | 12 | 31 (18 to 47) | 40 | 8 | 20 (10 to 35) | 41 | 16 | 39 (25 to 55) | −5 (−18 to 9) 0.509 | NA | |
| Birth attendant was able to correctly manage newborn resuscitation | Intervention | 55 | 6 | 11 (5 to 23) | 55 | 12 | 22 (13 to 35) | 57 | 27 | 47 (35 to 60) | 57 | 27 | 47 (35 to 60) | 36 (19 to 54) <0.001 | 0.038 | 0.775 |
| | Comparison | 37 | 5 | 14 (6 to 29) | 36 | 5 | 14 (6 to 30) | 36 | 8 | 22 (11 to 39) | 38 | 9 | 24 (13 to 40) | 10 (−8 to 29) 0.276 | NA | |
| Newborn bathing was delayed until at least 24 hours after birth | Intervention | 519 | 226 | 44 (36 to 52) | 492 | 224 | 46 (37 to 54) | 511 | 305 | 60 (50 to 69) | 458 | 285 | 62 (52 to 71) | 18 (8 to 30) 0.002 | 0.012 | 0.034 |
| | Comparison | 492 | 161 | 33 (26 to 40) | 435 | 136 | 31 (24 to 39) | 480 | 188 | 39 (30 to 49) | 412 | 169 | 41 (32 to 51) | 8 (−1 to 18) 0.090 | NA | |
| Newborn received clean cord care | Intervention | 519 | 216 | 42 (34 to 49) | 494 | 212 | 43 (36 to 50) | 511 | 331 | 65 (57 to 72) | 458 | 335 | 73 (66 to 79) | 31 (21 to 42) <0.001 | <0.001 | <0.001 |
| | Comparison | 492 | 140 | 28 (23 to 35) | 438 | 134 | 31 (24 to 38) | 480 | 159 | 33 (26 to 41) | 413 | 208 | 50 (41 to 59) | 22 (13 to 31) <0.001 | 0.028 | |
| Newborn was breastfed within 1 hour of birth | Intervention | 511 | 270 | 53 (46 to 60) | 489 | 283 | 58 (49 to 66) | 509 | 531 | 69 (62 to 75) | 454 | 304 | 67 (59 to 74) | 14 (3 to 25) 0.016 | 0.028 | 0.524 |
| | Comparison | 483 | 215 | 45 (39 to 50) | 430 | 207 | 48 (40 to 56) | 474 | 262 | 55 (49 to 62) | 409 | 249 | 61 (51 to 70) | 16 (7 to 26) 0.002 | 0.586 | |
| **Postpartum and postnatal care period** | | | | | | | | | | | | | | | | |
| Women received a postpartum check within 2 days of delivery | Intervention | 519 | 56 | 11 (7 to 17) | 492 | 39 | 8 (5 to 12) | 507 | 59 | 12 (8 to 17) | 457 | 60 | 13 (9 to 19) | 2 (−4 to 9) 0.451 | NA | 0.018 |
| | Comparison | 492 | 24 | 5 (3 to 9) | 434 | 23 | 5 (3 to 8) | 480 | 11 | 2 (1 to 5) | 413 | 34 | 8 (5 to 13) | 3 (1 to 8) 0.002 | 0.002 | |
| Newborns received a postnatal check within 2 days of birth | Intervention | 519 | 28 | 5 (3 to 9) | 491 | 27 | 5 (3 to 10) | 510 | 32 | 6 (4 to 10) | 458 | 31 | 7 (4 to 11) | 2 (−2 to 5) 0.391 | NA | 0.059 |
| | Comparison | 492 | 18 | 4 (2 to 7) | 435 | 10 | 2 (1 to 4) | 479 | 5 | 1 (0.3 to 3) | 412 | 17 | 4 (2 to 10) | 0 (−4 to 4) 0.825 | NA | |
| Newborns with suspected sepsis treated with antibiotics | Intervention | 43 | 15 | 35 (22 to 50) | 33 | 13 | 39 (23 to 58) | 25 | 9 | 36 (15 to 64) | 15 | 11 | 73 (44 to 91) | 38 (12 to 65) 0.006 | 0.198 | 0.273 |
| | Comparison | 29 | 13 | 45 (28 to 62) | 23 | 9 | 39 (24 to 57) | 20 | 4 | 20 (9 to 38) | 21 | 10 | 48 (26 to 70) | 3 (−27 to 33) 0.852 | NA | |

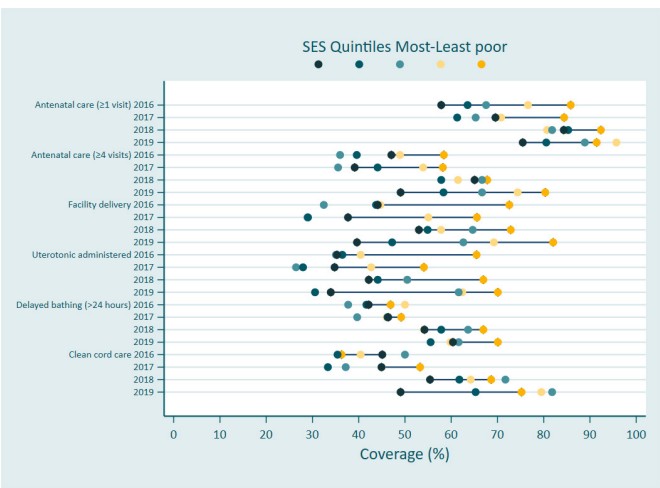

**Figure 3** Equiplot of change in coverage 2016–2019 by household wealth quintile. SES, socioeconomic status.

Coverage of timely postnatal and postpartum care for both home and facility births remained persistently low. This challenge is not unique to Gombe state, with very low coverage shown in other states of Nigeria,[25] and the sub-Saharan African region more widely.[47] Commonly reported barriers were present, for example, suboptimal demand generation, inadequate staffing and workload challenges, and multiple sociocultural barriers.[45 46] However, the power of the partnership to address other apparently intractable problems was not harnessed in relation to timely postnatal and postpartum care visits, and actors did not conceive successful mechanisms that could address the problem in their setting. The role of community-based ward development committees, whose members include fathers, husbands and community leaders, had an important role in demand-generation, advocacy and accountability in this setting. NGO partners aimed to leverage the power and influence of ward committees for postnatal health, but these were enacted towards the end of our study period. Nonetheless, we suggest that such engagement is important for shifting norms and expectations where cultural barriers may restrict new mothers from returning to facilities soon after birth or accepting non-familial visitors during the immediate postpartum period.[48 49]

The Gombe Partnership for Maternal and Newborn Health achieved greater success than other recent NGO driven attempts and this study suggests the importance of the nature of the partnership and the mechanisms of implementation. The partnership was government-led with priorities and implementation guided by policy priorities.[50 51] This engendered political will. Further, good leadership, governance and accountability facilitated successful implementation despite a change in government during the 2019 general elections, which fell six months before the final data collection period of this study. The importance of political engagement and leadership has been highlighted by many, including the extensive work of Gilson and Agyepong.[52] From this foundation

**Table 5** Continued

| Indicator | Area | 2016 N | 2016 n | 2016 % (95% CI) | 2017 N | 2017 n | 2017 % (95% CI) | 2018 N | 2018 n | 2018 % (95% CI) | 2019 N | 2019 n | 2019 % (95% CI) | Percentage point difference 2019–2016 (95% CI) P value* | Linear trend P value† | P value interaction time and area‡ |
|---|---|---|---|---|---|---|---|---|---|---|---|---|---|---|---|---|
| Facilities had newborn sepsis treatment available | Intervention | 55 | 20 | 36 (25 to 50) | 55 | 38 | 69 (56 to 80) | 57 | 36 | 63 (50 to 75) | 57 | 38 | 67 (53 to 78) | 31 (10 to 50) 0.004 | <0.001 | 0.054 |
| | Comparison | 39 | 12 | 31 (18 to 47) | 39 | 15 | 38 (24 to 55) | 40 | 9 | 23 (12 to 38) | 41 | 11 | 27 (15 to 43) | −4 (−22 to 14) 0.663 | NA | |

Models of household survey indicators adjusted for maternal education. Denominators vary due to missing data.
*P value based on t-test.
†Test for linear trend based on likelihood ratio test.
‡P value for interaction between area (intervention) and survey year (2019).
ANC, antenatal care; CI, Confidence interval.

of cooperation, the intervention components and implementation plan were built to achieve a common goal. From the inception, a plan for sustainability and transition from NGO implementation and donor-financing was in place.[24] A substantial focus on community participation and accountability was also included, principally through the role of ward development committees.

Implementation was pragmatic and a strong emphasis was placed on adoption into the health system.[20] The government and partners met six monthly to examine monitoring and evaluation data, assess progress, highlight challenges and plan for improvements. A recent systematic review has reported that many of these approaches, including a process of data-based feedback and adaptation have been important in the vertical scale-up and institutionalisation of public health interventions.[53]

### Strengths and limitations

This was a large and adequately-powered study which drew on multiple data sources and time points, and consistently applied globally recommended measurement standards. However, there were three noteworthy limitations. First, as for all maternal and newborn health data collected through maternal self-report, there was the possibility of recall bias, despite our relatively short 12-month recall period. Further, we cannot discount the possibility that some women were not able to accurately report on all questions.[54] Nonetheless, we would not expect recall bias or poor validity to be differential between groups or between surveys. Second, the packaging of interventions means that successful outcomes cannot be attributed to a single intervention, rather we can only conclude that the interventions in this health system strengthening package collectively contributed to the effects observed. Third, selection of intervention areas by government was pragmatic and baseline characteristics between areas were not balanced. Despite adjustment in multivariate models and an analysis strategy that adjusted for baseline starting points, some residual confounding by area may remain, potentially overestimating effects of the intervention. The selection of intervention areas and the inability to build in buffer zones also resulted at times in close geographical boundaries between comparison and intervention areas. This may have enabled spillover of effects created by women seeking care and using services from a facility that did not correspond to the intervention status of the ward in which she lived. This could have underestimated effects.

### CONCLUSIONS

After four years, this government-led programme achieved improvements in life-saving interventions for mothers and newborns, demonstrating that even in settings where resources are constrained, the needle can be shifted in the right direction. In this example, multiple interventions that spanned different health system pillars were implemented together as a package, and it is plausible that addressing demand and supply problems simultaneously was important. But we conclude that essential and reproducible elements of this programme lie not for the most part in the individual components of the intervention, but in the way the programme was designed and implemented, with its focus on government leadership and strong stakeholder partnership.

**Author affiliations**
[1]London School of Hygiene & Tropical Medicine, London, UK
[2]Society for Family Health, Abuja, Nigeria
[3]Pact, Abuja, Nigeria
[4]Data Research and Mapping Consult Limited, Abuja, Nigeria
[5]Office of the Honourable Commissioner for Health (and former Executive Secretary GSPCDA), Gombe State Ministry, Gombe, Nigeria
[6]Office of the Executive Secretary, Gombe State Primary Health Care Development Agency, Gombe, Nigeria

**Acknowledgements** We are grateful to the women and families who participated and shared their experiences. We would like to thank all the health facility participants and birth attendants who took the time to contribute to our study. We would like to acknowledge Inuwa Jalingo from Data Research and Mapping Consult and recognise the contribution of Data Research and Mapping Consult field staff and supervisors who conducted the surveys. We would like to acknowledge the contribution of Vincent Ahonsi, Ester Agboa and Tunde Segun from Evidence for Action, as well as Wale Adeleye from Champions for Change. Sincere thanks to all collaborators at the Gombe State Primary Health Care Development Agency, Gombe State Ministry of Health, Society for Family Health Nigeria, Pact Nigeria, Champions for Change and Evidence for Action Mamaye. We would also like to acknowledge the contribution of our respected colleague and team member, Deepthi Wickremasinghe from the London School of Hygiene & Tropical Medicine, who sadly died on 2 April 2020. Her wisdom, kindness and dedication to improving maternal and newborn health are greatly missed.

**Contributors** BW conducted the analysis and drafted the manuscript. EB and EA provided statistical advice. BW, EB and JE verified the underlying data. NU, EB, AAB, JE, KM, JS, NS and TM all provided important intellectual input. JA, ABB, MO, RS, AG and AS coordinated study implementation and contributed to country data interpretation. UAU coordinated data collection activities and contributed to country data interpretation. TM and JS conceived the study and are guarantors. All authors have read, critically reviewed the manuscript and approved the final version of the manuscript.

**Funding** The research was supported by IDEAS—Informed Decisions for Actions to improve maternal and newborn health (http://ideas.lshtm.ac.uk), which is funded through a grant from the Bill & Melinda Gates Foundation (Gates Global Health Grant Number: INV-007644) to the London School of Hygiene & Tropical Medicine.

**Disclaimer** The funder of this study had no role in the study's design or conduct, data collection, analysis or interpretation of results, writing of the paper or decision to submit for publication. The corresponding author had access to all study data and responsibility for the decision to submit the paper for publication.

**Competing interests** None declared.

**Patient and public involvement** Patients and/or the public were not involved in the design, or conduct, or reporting, or dissemination plans of this research.

**Patient consent for publication** Not applicable.

**Ethics approval** This research was conducted with approval from: The Nigerian National Health Research Ethics Committee (NHREC/01/01/2007); the State Ministry of Health Gombe State (ADM/S/658/Vol. II/66); and the London School of Hygiene & Tropical Medicine (12181). The free and informed written consent of all participants was obtained.

**Provenance and peer review** Not commissioned; externally peer reviewed.

**Data availability statement** Data are available upon reasonable request. De-identified participant data sets, protocols and questionnaires are uploaded to the LSHTM Data Repository. Data are available on reasonable request from the principal investigator of the IDEAS project and coauthor for this manuscript, TM (ORCID id 0000-0002-4228-4334). Reuse permitted on request.

**ORCID iDs**
Barbara Willey http://orcid.org/0000-0002-5392-3357
Nasir Umar http://orcid.org/0000-0001-5119-0092
Emma Beaumont http://orcid.org/0000-0001-5763-0933
Elizabeth Allen http://orcid.org/0000-0002-2689-6939
Antoinette Bhattacharya http://orcid.org/0000-0001-5400-9383
Josephine Exley http://orcid.org/0000-0002-6501-0854
Krystyna Makowiecka http://orcid.org/0000-0003-1075-9697
Joanna Schellenberg http://orcid.org/0000-0002-0708-3676
Neil Spicer http://orcid.org/0000-0003-3451-8003
Tanya Marchant http://orcid.org/0000-0002-4228-4334

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
