## [Reviewer comments · BMJ Open]

ARTICLE DETAILS

TITLE (PROVISIONAL)	Improving maternal and newborn health services in northeast Nigeria through a government-led partnership of stakeholders: a quasi-experimental study
AUTHORS	Willey, Barbara; UMAR, NASIR; Beaumont, Emma; Allen, Elizabeth; Anyanti, Jennifer; Bello, Abubakar Bala; Bhattacharya, Antoinette Alas; Exley, Josephine; Makowiecka, Krystyna; Okolo, Magdalyn; Sani, Rabi; Schellenberg, Joanna; Spicer, Neil; Usman, Umar Adamu; Gana, Ahmed; Shuaibu, Abdulrahman; Marchant, Tanya

VERSION 1 – REVIEW

REVIEWER	Srivastava, Ashish JHPIEGO
REVIEW RETURNED	05-Apr-2021

GENERAL COMMENTS	This is a well written manuscript which describes the results of a large scale evaluation of a government led intervention to improve MNH care. The manuscript shares important findings and learnings, which will be of interest to public health professionals, specifically to those from low and middle income countries. I have reviewed the manuscript and am sharing a few suggestions/observations, which I hope will help the authors to bring in more clarity. Introduction: 1. Details of the intervention per say needs to be added. The authors have summarized the components in a figure, however they do not elaborate on them in the manuscript. The authors assert that the intervention was government led and aimed at coordinating efforts of different stakeholders/actors, but do not delve into what specifically these efforts were. More details of the Gombe partnership for MNH needs to be added. Who all were the stakeholders? What specific components did they support/implement? More details of the different components of the intervention summarized in figure 1 needs to be added. What mechanisms were put in place for ensuring improved co-ordination between different partners? All we know is that the partners implemented a package of evidence based interventions spanning across the six WHO health systems pillars.2. Conclusion: One could also argue that different co-interventions (as summarized in figure 1), spanning across different WHO health systems pillars, being implemented together could have made the difference.3. A section elaborating on the different study tools, their components/domains covered needs to be added. More information regarding pre-testing needs to be added. On how many respondents/facilities was the pre-testing done, what were
--

	the key findings and what changes were made to the tool/s. How much time did it take to administer each of the utilized tools? 4. Who were the trained interviewers? What was their background? How long were they trained? Was data collection supervised? Who collected facility readiness data? Were they the same trained interviewers? 5. Page 7, Line 23 – 25, Line 35-36: On sample size being sufficient, please elaborate on the baseline ranges for which the sample size is sufficient for the expected change. 6. Page 7, line 32-33: 'one PHC selected at random from each of the 40 comparison wards'. Did the team select 40 clusters or 40 wards from comparison area? Please clarify on the relation between ward and sampling cluster. 7. Page 7, Line 53-55 and Page 8, Line 8-9: Same information is repeated twice. Can delete it at one place. 8. Page 10, line 25-26: Was it written or verbal consent? 9. Were there women (in the sample) who availed MNH services from private healthcare facilities or higher level health care facilities? Were they included in the survey? How was their data analyzed? 10. What is the basis for including receipt of blood test, BP measurement and Urine test as a measure of quality ANC? Weren't questions around respectful maternal care, early identification and referral of high risk cases etc. included? 11. Were medical records/ANC cards of women reviewed to validate information provided by the interviewed recently delivered women? 12. Page 10, Line 56-58: Please correct the typo in the numerator for "New born received clean cord care". 13. Please clarify on the indicator for "Newborns with suspected sepsis treated with antibiotics" – Was this not asked to recently delivered women, like all other newborn related indicators? 14. How was receiving post-partum or post-natal check defined? Was a certain cadre of the staff expected to do these checkups? 15. What was the cadre of birth attendants who were assessed for their capacity to manage pre-eclampsia or NBR? Did it vary significantly between intervention and comparison facilities? 16. Table 4: Please check the p values for "Women received post-partum check within 2 days of delivery". 17. The survey respondents were asked to recall details of an event which may have happened almost a year back. This could have introduced a recall bias. More so, information like receiving a uterotonic or undergoing the syphilis test are details that a woman may not be able to comment upon/recall and it would be more appropriate to validate it from medical records. Please acknowledge these points in the limitations section.
--	---

REVIEWER	Kouanda, Seni Institut de Recherche en Sciences de la Santé
REVIEW RETURNED	06-May-2021

GENERAL COMMENTS	The paper is well writing and have the potential to inform the policy to the impact of the interventions. however, it's remain some issues. Minor comments  - Delete the comma in line 23 (page 4). - What was a state-wide components (line 37, page 5)? Not defined in text. - What was the rationale behind the number of facility and birth attendant surveys? (line 14 to 24, page 7)
---

	- Is the sample size powered for each cross sectional study? If so, please indicate it (page 7). - A map describing the study site would be useful for the description of the sites. Major comments - We need the details of the content of each intervention (perhaps as additional file) - Table 4 is difficult to read. One figure for each group of indicators would make the data easier to read (page 16). - The main limitation of this study is that the groups were not randomly selected. This limitation is not sufficiently discussed in the article. It is not clear what effect it might have on the results presented by the authors (line 21, page 22).
--	---

VERSION 1 – AUTHOR RESPONSE

Reviewer: 1		
Dr. Ashish Srivastava, JHPIEGO		
Introduction:		
1. Details of the intervention per say needs to be added. The authors have summarized the components in a figure, however they do not elaborate on them in the manuscript. The authors assert that the intervention was government led and aimed at coordinating efforts of different stakeholders/actors, but do not delve into what specifically these efforts were. More details of the Gombe partnership for MNH needs to be added. Who all were the stakeholders? What specific components did they support/implement? More details of the different components of the intervention summarized in figure 1 needs to be added. What mechanisms were put in place for ensuring improved co-ordination between different partners? All we know is that the partners implemented a package of evidence based interventions spanning across the six WHO health systems pillars.	Thank you for this helpful comment, we have made substantial edits and incorporated additional detail.	p.5 & 6
2. Conclusion: One could also argue that different co-interventions (as summarized in figure 1), spanning across	Thank you, we have now included this point in the discussion.	p.23

different WHO health systems pillars, being implemented together could have made the difference.		
3. A section elaborating on the different study tools, their components/domains covered needs to be added. More information regarding pre-testing needs to be added. On how many respondents/facilities was the pre-testing done, what were the key findings and what changes were made to the tool/s. How much time did it take to administer each of the utilized tools?	Thank you for this suggestion which has led to substantial editing of the manuscript text describing the surveys.	p.9
4. Who were the trained interviewers? What was their background? How long were they trained? Was data collection supervised? Who collected facility readiness data? Were they the same trained interviewers?	We have now added additional detail, please see comment 3 above.	p.9
5. Page 7, Line 23 – 25, Line 35-36: On sample size being sufficient, please elaborate on the baseline ranges for which the sample size is sufficient for the expected change.	Thank you, we have elaborated on the sample size calculation	p.8
6. Page 7, line 32-33: 'one PHC selected at random from each of the 40 comparison wards'. Did the team select 40 clusters or 40 wards from comparison area? Please clarify on the relation between ward and sampling cluster.	Thank you for this question. 40 clusters from each of the intervention and comparison area were sampled, 80 in total. A ward is a sub-LGA (local governmental area- equivalent to district) political enumeration area, defined by the National Population Commission. In Gombe State at the time of the study there were 11 LGAs and 114 wards. Cluster sampling was performed by listing all enumeration areas (wards), cumulating their population size, and systematically selecting areas with probability proportional to size. All households in selected enumeration areas were listed, and enumeration areas segmented into groups of 75 or fewer households. During the second stage, field teams randomly selected one segment from each enumeration area as the cluster to be surveyed.	
7. Page 7, Line 53-55 and Page 8, Line 8-9: Same information is repeated twice. Can delete it at one place.	We have re-written substantial sections, and believe we have now removed duplications.	

8. Page 10, line 25-26: Was it written or verbal consent?	We have now clarified written consent, thank you	p.9
9. Were there women (in the sample) who availed MNH services from private healthcare facilities or higher level health care facilities? Were they included in the survey? How was their data analyzed?	The household survey enrolled all women of reproductive age who lived in sampled clusters. As such there were no restrictions based on type of facility utilised: that is, the coverage estimates are population level estimates. In the context of Gombe State, maternal and newborn health is predominantly accessed at public primary level healthcare facilities. For example, of the 870 women providing data about location of their most recent delivery, just 67 - 8% - reported accessing a hospital.	
10. What is the basis for including receipt of blood test, BP measurement and Urine test as a measure of quality ANC? Weren't questions around respectful maternal care, early identification and referral of high risk cases etc. included?	The evaluation tools were developed to reflect the global and national guidelines available to implementers at the inception of the project so as to present a fair measure of inputs from baseline. The developments in recommendations for best practice in maternal and newborn health, including respectful care, emerged after the inception of this project and so have not been included as core to this effectiveness evaluation. However, the team has conducted special studies to explore emerging themes where possible, for example see https://pubmed.ncbi.nlm.nih.gov/32201626/ and https://pubmed.ncbi.nlm.nih.gov/33099494/	
11. Were medical records/ANC cards of women reviewed to validate information provided by the interviewed recently delivered women?	Thank you for this comment. Mother's health cards were requested during the household survey and interviewers were trained to check, and prompted to check during the survey whether information provided by mothers matched the records on her health card. However, consistent with chronic stock-outs of health cards in the study area, only a minority of women were able to produce one: for example, in 2016 just one quarter of women with a recent birth were able to produce an antenatal card and many of these were informal records or notebooks. As such interviewers relied on women's self-report. As per comment 17 below, we include this as a possible limitation in the manuscript text.	p.23
12. Page 10, Line 56-58: Please correct the typo in the numerator for "New born received clean cord care".	Now corrected, thanks	
13. Please clarify on the indicator for "Newborns with suspected sepsis treated with antibiotics" – Was this not asked to recently delivered women, like	All women with a recent delivery were included in module three of the household survey. However, within module three only women whose baby had been born within 60 days of the survey were asked about treatment seeking for an ill newborn.	

all other newborn related indicators?		
14. How was receiving post-partum or post-natal check defined? Was a certain cadre of the staff expected to do these checkups?	A post-partum or post-natal check was defined as receiving a visit from a trained VHW or formal health care provider to check on health status after birth, within the specified time-frame. Obtained through the household survey by self-report by women (or care-giver for newborns).	
15. What was the cadre of birth attendants who were assessed for their capacity to manage pre-eclampsia or NBR? Did it vary significantly between intervention and comparison facilities?	Thank you for this comment. We have amended the text of the results to include information about the birth attendants interviewed in our sample. These were principally CHEWs or non-clinical staff, with no differences in cadre between areas.	p.16
16. Table 4: Please check the p values for “Women received post-partum check within 2 days of delivery”.	We have re-run the models and checked this and it is correct. There are large numbers in the model overall as it includes all survey years (3794), but interpretation is obviously no improvement in either area over time and extremely low coverage.	
17. The survey respondents were asked to recall details of an event which may have happened almost a year back. This could have introduced a recall bias. More so, information like receiving a uterotonic or undergoing the syphilis test are details that a woman may not be able to comment upon/recall and it would be more appropriate to validate it from medical records. Please acknowledge these points in the limitations section.	Thank you for this comment. We restricted our analysis of births to events in the 12 months before survey, resulting in a much shorter recall period than the MICS or DHS survey protocols which typically require 3-5 years recall. Nonetheless, we cannot discount the possibility that some women were not able to accurately recall and report on all questions, although we would not expect recall bias to be differential between groups or between surveys [please see https://pubmed.ncbi.nlm.nih.gov/31360449/]	p.23
Reviewer: 2		
Prof. Seni Kouanda, Institut de Recherche en Sciences de la Santé, Institut Africain de santé publique		
Minor comments		
- Delete the comma in line 23 (page 4).	Now deleted, thanks	p.4
- What was a state-wide components (line 37, page 5)? Not defined in text.	Now defined	p.6

What was the rationale behind the number of facility and birth attendant surveys? (line 14 to 24, page 7)	Thank you for your question. Household surveys were conducted at annual intervals which made sense given that we were analysing outcomes for women with births in the previous 12 months. However, it was felt that facility-level indicators might be amenable to more frequent change and government was keen to receive more regular updates, both to track progress and to inform their decision making. To facilitate this request it was agreed to increase the frequency of facility surveys to six-monthly throughout the engagement period.	
- Is the sample size powered for each cross sectional study? If so, please indicate it (page 7).	Thank you, we have now clarified	p.8
- A map describing the study site would be useful for the description of the sites.	Thank you for your suggestion, unfortunately a detailed map of the study site is not available, partly as an ethical consideration to protect the anonymity of study facilities and communities.	
Major comments		
- We need the details of the content of each intervention (perhaps as additional file)	Thank you, we have included this now and have made substantial edits and incorporated additional detail	p.5
13. Please clarify on the indicator for “Newborns with suspected sepsis treated with antibiotics” – Was this not asked to recently delivered women, like all other newborn related indicators?	Thank you for this comment. Summarising large amounts of information is always a challenge and while we acknowledge the ease of visual interpretation that a figure would bring, we feel that this is not outweighed by the loss of detail and information of the sample sizes and error estimates. Should the manuscript be accepted for publication, we would be willing to continue this discussion with the typesetters and include figures as additional appendices.	
The main limitation of this study is that the groups were not randomly selected. This limitation is not sufficiently discussed in the article. It is not clear what effect it might have on the results presented by the authors (line 21, page 22).	One consequence of our plausibility, non-randomised, study design is that the effects observed may be an under or over estimation. In this study of 'real-world' action, we were unable to influence site selection and so designed a widely applied plausibility design to accommodate government pragmatism. This is included as a limitation of the study (where we have further elaborated on study limitations, as raised by both reviewers).	p.23